# Deficiency of GABARAP but Not Its Paralogs Causes Enhanced EGF-Induced EGFR Degradation

**DOI:** 10.3390/cells9051296

**Published:** 2020-05-22

**Authors:** Jochen Dobner, Indra M. Simons, Kerstin Rufinatscha, Sebastian Hänsch, Melanie Schwarten, Oliver H. Weiergräber, Iman Abdollahzadeh, Thomas Gensch, Johannes G. Bode, Silke Hoffmann, Dieter Willbold

**Affiliations:** 1Institut für Physikalische Biologie, Heinrich-Heine-Universität Düsseldorf, 40225 Düsseldorf, Germany; j.dobner@fz-juelich.de (J.D.); i.simons@fz-juelich.de (I.M.S.); 2Institute of Biological Information Processing: Structural Biochemistry (IBI-7), Forschungszentrum Jülich, 52425 Jülich, Germany; m.schwarten@fz-juelich.de (M.S.); o.h.weiergraeber@fz-juelich.de (O.H.W.); i.abdollahzadeh@fz-juelich.de (I.A.); si.hoffmann@fz-juelich.de (S.H.); 3Department of Gastroenterology, Hepatology and Infectiology, University Hospital, Heinrich-Heine-Universität Düsseldorf, 40225 Düsseldorf, Germany; rufinats@hhu.de (K.R.); Johannes.Bode@med.uni-duesseldorf.de (J.G.B.); 4Department of Biology, Center for Advanced Imaging (CAi), Heinrich-Heine-Universität Düsseldorf, 40225 Düsseldorf, Germany; Sebastian.Haensch@uni-duesseldorf.de; 5Institute of Biological Information Processing: Molecular and Cell Physiology (IBI-1), Forschungszentrum Jülich, 52425 Jülich, Germany; t.gensch@fz-juelich.de

**Keywords:** EGFR, GABARAP, receptor trafficking, degradation, Atg8, genome editing

## Abstract

The γ-aminobutyric acid type A receptor-associated protein (GABARAP) and its close paralogs GABARAPL1 and GABARAPL2 constitute a subfamily of the autophagy-related 8 (Atg8) protein family. Being associated with a variety of dynamic membranous structures of autophagic and non-autophagic origin, Atg8 proteins functionalize membranes by either serving as docking sites for other proteins or by acting as membrane tethers or adhesion factors. In this study, we describe that deficiency for GABARAP alone, but not for its close paralogs, is sufficient for accelerated EGF receptor (EGFR) degradation in response to EGF, which is accompanied by the downregulation of EGFR-mediated MAPK signaling, altered target gene expression, EGF uptake, and EGF vesicle composition over time. We further show that GABARAP and EGFR converge in the same distinct compartments at endogenous GABARAP expression levels in response to EGF stimulation. Furthermore, GABARAP associates with EGFR in living cells and binds to synthetic peptides that are derived from the EGFR cytoplasmic tail in vitro. Thus, our data strongly indicate a unique and novel role for GABARAP during EGFR trafficking.

## 1. Introduction

The epidermal growth factor receptor (EGFR/ErbB1) is a plasma membrane bound receptor tyrosine kinase (RTK) that is expressed in many different cell types and plays an important role in numerous processes, such as development, tissue homeostasis, and regeneration [1,2], by binding a variety of ligands, including transforming growth factor-α (TGF α) [3], amphiregulin [4], and the eponymous epidermal growth factor [5]. The binding of these ligands causes either homo- or heterodimerization with the other members of the erythroblastosis oncogene B (ErbB) superfamily ErbB2, ErbB3, and ErbB4 [6], leading to intrinsic kinase activation and autophosphorylation of distinct tyrosine residues in the C-terminal cytoplasmic region of the receptor [7,8]. The subsequent recruitment and activation of various downstream signaling pathways causes a plethora of cellular effects, depending on which pathway is activated, including cell growth, proliferation, differentiation, and motility [9,10,11]. Dephosphorylation and/or degradation of the activated receptor are necessary to strictly control and regulate these signaling events, thus preventing the sustained activation and uncontrolled cell growth that are found in many types of cancer [12,13].

Ligand-associated EGFR undergoes rapid internalization [14,15], which can either be clathrin-dependent [16] or independent [17]. Whereas the former mainly occurs under low ligand concentrations, leading to sustained EGFR signaling and enhanced recycling back to the plasma membrane through a non-degradative sorting pathway, with the latter being accompanied by monoubiquitination at several sites and packaging of activated receptors in intraluminal vesicles (ILV) of multivesicular bodies (MVB), followed by maturation to late endosomes and ultimately fusion with the lysosome where the receptor is degraded by pH-dependent hydrolases [18]. 

Autophagy is an intracellular degradation pathway [19]. Upon stress conditions, such as nutrient starvation, but also in response to oxidative stress and pathogen infection [20], membrane cisternae in the cytosol of cells arise and engulf cargo either non-selectively or selectively. Their closure finally yields double-membrane vesicles, termed autophagosomes. Ultimately, their fusion with the lysosome leads to the degradation of autophagosomal content [21]. This enables cells to survive by repurposing amino acids and other resources, to get rid of damaged organelles [22], but also to eliminate pathogens [23]. The autophagic degradation of RET (receptor tyrosine kinase Proto-oncogene tyrosine-protein kinase receptor) and associated proto-oncogene tyrosine-protein kinase SRC via autophagy has also been reported [24,25]. Apart from degradation, components of the autophagic machinery are also implicated in the secretion processes by facilitating a form of unconventional protein secretion [26,27]. Proteins of the autophagy-related (Atg) protein family have first been identified in yeast and they are involved in every step of the autophagic process [28,29]. While, in yeast only a single *Atg8* gene exists [30], in mammalian cells the family has expanded into a number of paralogs [31]. The microtubule-associated proteins 1A/1B light chain 3 (LC3) proteins A, B, and C are grouped in the LC3 subfamily, whereas γ-aminobutyric acid type A receptor-associated protein (GABARAP) and its two paralogs GABARAPL1 and GABARAPL2 form the GABARAP subfamily, according to their degree of relation. Besides (canonical) autophagy, GABARAP subfamily members have been described to play pivotal roles in many cellular processes, such as immunity, receptor trafficking, unconventional secretion of leaderless proteins [32,33,34], and interaction with viral proteins [35,36,37]. However, because they share high sequence and structural similarity [38] within and between subfamilies, the elucidation of their exact and especially non-redundant functions requires the development of highly specific and sensitive readout systems. Progress towards this goal has been made in the field of autophagy, especially regarding their roles during autophagosome biogenesis (e.g., [39,40,41]) as well as selective cargo loading via cargo receptor interaction ([42,43,44]). Respective overviews can be found in several recent reviews (e.g., [32,34,45,46,47,48]). The direct binding of interaction partners to Atg8 proteins is mediated by a canonical interaction motif, generally known as LC3-interacting region (LIR) or GABARAP interaction motif (GIM) in the case of GABARAP subfamily ligands [49], which can reach various levels of specificity [50]. Very recently, an additional motif, related to the ubiquitin interacting motif (UIM), was described utilizing a binding region localized opposite to the LIR/GIM-docking site on the Atg8 protein surface [51]. 

Additionally, it has long been known that the proteins of the GABARAP subfamily are involved in the regulation of cell surface receptor trafficking. GABARAP was first described to be associated to the name-giving GABAA receptor [52] and implicated in its trafficking [53]. It was also described to be associated with the Transferrin receptor [54] and be important in the clustering of Transient receptor potential cation channel subfamily V member 1 (TRPV1) at the cell surface [55]. Furthermore, angiotensin II type 1 (AT1) receptor plasma membrane expression was described to be mediated by GABARAP [56], while sodium-dependent phosphate transport protein 2A (SLC34A1) levels were found to be increased in its absence [57]. Recently, GABARAPL2 was reported to be directly involved in regulating the protein levels of Parkin associated endothelin like receptor (PAELR) [58]. GABARAPL1, in turn, has also been described to be implicated in trafficking of the GABAA receptor [59] and the κ-opioid receptor [60]. Importantly, GABARAPL1 has already been connected with increased EGFR surface expression under hypoxic conditions without altering the total EGFR levels [61]. However, in almost all above-mentioned autophagy-unrelated functions, systematic analysis revealing unique and non-redundant roles of the three human GABARAP subfamily members are largely lacking.

Therefore, the aim of this work was to analyze the role of the different members of the GABARAP subfamily of human Atg8 family proteins in trafficking, signaling, and degradation of the cell surface receptor EGFR as a model RTK.

## 2. Materials and Methods

### 2.1. Materials

A list of antibodies (Table A1) and RT-PCR primers (Table A2) used in this study can be found in Appendix B. Unless stated otherwise, antibodies were used at dilutions according to the manufacturer’s instructions.

### 2.2. Cell Culture

Human hepatoma Huh7.5 cells [62] were maintained in Dulbecco’s Modified Eagle Medium (DMEM) high glucose (F0445, Biochrom, Berlin, Germany) that was supplemented with 10% (*v*/*v*) heat-inactivated fetal bovine serum (FBS, 10270-106, ThermoFisher Scientific, Waltham, MA, USA), 2 mM L-glutamine (25040081, ThermoFisher Scientific), 1% penicillin/streptomycin (15140-122, ThermoFisher Scientific), 10 µL/mL non-essential amino acids (NEAA, 11140-035, ThermoFisher Scientific) at 37 °C and 5% CO_2_. The use of the Huh7.5 cell line is covered by a material transfer agreement with Apath, L.L.C. (New York City, NY, USA). Human embryonic kidney 293 (HEK293) Flp-In T-Rex cells [63,64] were maintained in DMEM high glucose that was supplemented with 10% (*v*/*v*) heat-inactivated FBS at 37 °C and 5% CO_2_. The cells were split regularly at 80% confluency and used for a maximum of 30 passages after thawing. Cells were routinely checked for mycoplasma contamination.

For treatment with EGF and inhibitors 0.14 × 10^6^ cells were seeded two days prior treatment in wells of six-well plates. The cells were pretreated with either 10 µM lactacystin (L6785, Sigma–Aldrich, St. Louis, MO, USA) for 1 h, 100 µM chloroquine (14774, Cell Signaling Technologies, Danvers, MA, USA) for 2 h or left untreated, followed by treatment with 40 ng/mL EGF (11376454001, Roche, Basel, Switzerland) for up to 180 min.

### 2.3. CRISPR/Cas9 Mediated Knockout Generation

Knockout (KO) cell lines were generated, as described [65]. In brief, the cells were transfected with KO plasmids based on pSpCas9(BB)-2A-GFP (PX458) [66], single sorted for fluorescent protein (FP) positive signals via fluorescence-activated cell-sorting (FACS) in wells of 96-well plates, clonal lines recovered and occurrence of genome editing verified via the amplification of a 400 bp product flanking the target site and Sanger sequencing as well as on protein level with specific antibodies. Knock-in (KI) of an enhanced green fluorescent protein (EGFP) at the endogenous locus of GABARAP was achieved by transfecting a homology-directed repair (HDR-) plasmid containing homology arms 1 kbp up- and downstream of the CRISPR target site flanking the sequence for EGFP. The cells were serum starved 24 h prior transfection to enhance HDR, sorted by FACS as single cells in wells of 96-well plates, and recovered and analyzed as the KO cells. The resulting cell line was termed GFP-GABARAP. Sequences for primers used for Sanger sequencing can be found in Appendix A.

### 2.4. Transient Transfection

Nucleofection was performed according to the manufacturer’s instructions (Lonza, Basel, Switzerland) with 4D Nucleofector. In brief, HEK293 Flp-In T-REx cells were split 2–3 days prior to nucleofection. On the day of nucleofection, 1 × 10^6^ cells were nucleofected with 2 μg plasmid DNA and cell membranes recovered by adding warm RPMI (21875034, ThermoFisher Scientific) medium containing 10% FBS for 10 min. at 37 °C to each reaction. Afterwards, the cells were seeded into wells of 12-well plates and treated with 40 ng/mL EGF for up to 180 min. two days post nucleofection.

Lipofection with Lipofectamin2000 reagent was performed according to the manufacturer’s instructions (11668019, Qiagen, Hilden, Germany). Briefly, one day prior transfection, 3 × 10^5^ Huh7.5 cells were seeded in wells of 6-well plates in growth medium without antibiotics. On the day of transfection, 3 µg of each plasmid DNA and 10.5 µL Lipofectamin2000 reagent were diluted in 150 µL Opti MEM (1×) + GlutaMAX reduced serum medium (51985042, ThermoFisher Scientific), mixed and incubated for 5 min. at RT to form plasmid-lipid complexes. Afterwards, 250 µL of this solution were added dropwise to wells of the 6-well plates in order to reach a final plasmid amount of 2.5 µg per well and then incubated for two days at 37 °C until further experiments. EGFR-GFP was a gift from Alexander Sorkin (Addgene plasmid #32751).

### 2.5. Immunoblot

The cells were washed once with ice cold PBS and then harvested by scraping into cell lysis buffer (1% Triton, 20 mM Tris/HCl, pH 7.4, 13.6 mM NaCl, 2 mM EDTA, 50 mM β-glycerophosphate, 20 mM sodium pyrophosphate, 1 mM Na3VO_4_, 4 mM benzamidine, 0.2 mM Pefabloc, 5 μg/mL aprotinin, 5 μg/mL leupeptin, 10% glycerol, and 0.2% SDS) on ice. Cell lysis was carried out by incubating samples on ice for 10 min. The supernatants were cleared by centrifugation at 20,000 g for 15 min. at 4 °C, transferred to fresh tubes, and protein concentration was determined by BCA assay. In general, 20–30 µg of whole cell protein lysate were supplemented with 4 × Lämmli buffer (250 mM Tris-HCl pH 6.8, 40% glycerol, 5% SDS, 0.005% bromophenol blue) supplemented with fresh 8% 2-mercaptoethanol, boiled for 7 min. at 95 °C and loaded on 10 or 12 %-PAA gels for SDS-PAGE. After SDS-PAGE, gels were imaged with BioRad Imager using the stain-free method in order to determine protein loading [67] and then transferred to 0.4 µm PVDF membrane via semidry blotting at 0.77 mA/cm² gel constant current for 1 h. The membranes were afterwards cut, washed for 5 min. with TBS containing 0.1% Tween20 (TBS-T) at RT, blocked with 5% BSA in TBS-T for 1 h at RT, and incubated with specific antibodies overnight at 4 °C. On the next day, blots were washed thrice with TBS-T at RT for 20 min. and incubated with fluorescently labeled secondary antibodies for 1 h at RT wrapped in aluminium foil. Afterward, membranes were washed thrice with TBS-T for 20 min. at RT. The protein levels were visualized either directly using BioRad Imager with detection settings specific for Alexa488 or Alexa647 in the case of fluorescent protein conjugated antibodies or after 2 min. incubation with Western Bright ECL spray (K-12049-D50, Advansta, San Jose, CA, USA) by chemiluminescence in case of HRP conjugated antibodies. The protein expression levels were normalized to total protein loading, as determined by stain-free analysis.

### 2.6. RT-PCR

The cells were washed once with PBS and harvested by scraping into cell lysis buffer RLT containing 0.01% 2-mercaptoethanol, followed by cell homogenization using Qiashredder spin columns (79656, Qiagen). The total RNA was extracted using RNeasy miniprep kit (74106, Qiagen). Concentration was determined by NanoDrop1000 and 1 µg total RNA was reverse transcribed into cDNA with Quantitect Reverse Transcription kit (205314, Qiagen) while using oligomeric (dT) primers and including a DNAse digestion step. The resulting cDNA was used in a 20 µL reaction mix containing 1 × SYBR green (A6002, Promega, Madison, WI, USA), 400 nM of each forward and reverse exon spanning specific primers, 1/250 diluted cDNA and RNAse free water. The mRNA levels were determined on Viia7 RT-PCR (ThermoFisher Scientific), normalized to *succinat dehydrogenase subunit a* (*SDHA*) as a reference gene, and expressed as fold-change compared to controls using the ΔΔCT method.

### 2.7. EGF Uptake Assay

For FACS-based EGF uptake assay, 0.07 × 10^6^ cells were seeded in wells of 12-well plates two days prior experiment. On the day of experiment, the cells were stimulated for up to 180 min. with Alexa647 labelled EGF (E35351, ThermoFisher Scientific). Afterwards, the cells were harvested by trypsin-EDTA treatment for 4 min. at 37 °C, resuspended in ice cold FACS buffer (PBS containing 2 mM EGTA, 1% FBS), washed twice with fresh FACS buffer, and analyzed with FACS Aria III (BD Bioscience, Franklin Lakes, NJ, USA). 

For pulse-based EGF uptake assay analyzed by immunofluorescence, Huh7.5 cells (2 × 10^5^) were seeded on fibronectin (F1141, Sigma–Aldrich, St. Louis, MO, USA) coated glass bottom µ-dishes (81158, ibidi, Martinsried, Germany) one day before analysis. The next day, Huh7.5 cells were precooled on ice for 10 min. Afterwards, cells were incubated in cold medium supplemented with 40 ng/mL EGF-Alexa647 conjugate (E35351, ThermoFisher Scientific) for 1 h at 4 °C to enable prebinding to surface EGFR. The medium was then replaced by full medium without EGF-Alexa647 and cells were incubated in at 37 °C for 5, 10, 30, 60, 120, or 180 min. or directly washed once with high salt and low pH buffer (0.2 M sodium acetate and 0.5 M NaCl, pH 4.5) to remove unspecific binding, fixed for 10 min. with 4% PFA in PBS, and washed twice with PBS. After each incubation time point, corresponding cells were treated the same way. 

For simultaneous immunofluorescence staining, fixed cells were permeabilized with 0.2% TritonX-100 in PBS for 30 min. at RT and then blocked with 1% bovine serum albumin (BSA, Sigma–Aldrich) at RT for 60 min. or overnight at 4–8 °C. Immunostaining was performed by incubation with primary antibodies for 60 min. at RT under gentle shaking. The cells were washed thrice for 5 min. with PBS followed by incubation with appropriate fluorescently labeled secondary antibody for 60 min. at RT in the dark, followed by two washing steps for 5 min. with PBS. 

### 2.8. Confocal Laser Scanning Microscopy

Images were acquired using ZEN black 2009 software operating a LSM 710 confocal laser scanning system (Carl Zeiss MicroImaging Inc., Dunedin, FL, USA) with a Plan-Apochromat 63x/1.40 Oil DIC M27 objective. The cell nuclei were stained with DAPI and measured in the 405 nm channel (MBS -405/760+). GFP-GABARAP was detected in the 488 nm channel (MBS 488), Transferrin (Tf)-Alexa 555 conjugate (T35352, Thermo Fisher Scientific) in the 543 nm channel (MBS 458/543) and EGF-Alexa 647 conjugate in the 633 nm channel (MBS 488/543/633), respectively. HEK293 GFP-GABARAP KI cells (2 × 10^5^) were seeded on fibronectin (F1141, Sigma–Aldrich) coated glass bottom µ-dishes (81158, ibidi), incubated overnight at 37 °C and 5% CO_2_ in growth medium. Directly before measuring, medium was replaced by cold medium containing 40 ng/mL EGF-Alexa 647. Data were post-processed using ImageJ [68] (version: 2.0.0-rc-43/1.50e).

### 2.9. EGFR Surface Expression Analysis

Plasma membrane proteins exhibiting extracellular primary amines were isolated with Pierce Cell Surface Protein Isolation Kit (89881, Thermo Fisher Scientific) according to the manufacturer’s instructions in order to analyze surface EGFR protein expression. In brief, two days prior labelling and isolation of plasma membrane proteins, 2 × 10^6^ cells per flask for a total of four T75 flasks per cell line were seeded for each biological replicate. On the day of isolation, each flask was quickly washed twice with ice cold PBS on ice and then labeled with Sulfo-NHS-Biotin shaking for 30 min. at 4 °C. The labelling reaction was then quenched, cells were scraped into solution, centrifuged, and lysed with lysis buffer for 30 min. with additional vortexing (every 5 min.) and sonication (every 8 min.) steps. Lysates were cleared and biotinylated proteins bound to NeutrAvidin beads for 1 h at RT with end-over-end mixing. The proteins were eluted with elution buffer containing 50 mM DTT for 1 h at RT with end-over-end mixing. Protein concentrations of eluates were determined by BCA and equal amounts were processed and loaded on 10% PAA gels for immunoblot analysis.

### 2.10. Spinning Disc Confocal Fluorescence Microscopy

During measurement, the cells were maintained in a StageTop Incubator system (Okolab S.R.L., Pozzuoli, Italy) on the microscope stage at 37 °C, 85% humidity, and 5% CO_2_. The spinning disk confocal fluorescence microscope (Acal BFI, Gröbenzell, Germany) is based on an inverted microscope (Eclipse Ti, Nikon Instruments Europe BV, Amsterdam, Netherlands) equipped with a multi-beam confocal scanner unit (CSU-W1, Yokogawa Deutschland GmbH, Ratingen, Germany) working according to the spinning disk confocal principle and two cw lasers for excitation at 488 nm (GFP-GABARAP) and 640 nm (EGF-Alexa647). The setup allows for fast (<50 ms) acquisition of confocal fluorescence images in three dimensions. Bright field images, as well as confocal fluorescence images with excitation at 488 and 640 nm, were recorded while using a 100 magnification oil immersion objective lens (CFI PLAN APOCHROMAT VC, NA = 1.40, Nikon) and appropriate multi-dichroic beamsplitter, an EMCCD camera (Ixon Ultra 897, Andor Technologies Ltd., Belfast, UK) as detector and an image splitting device (Optosplit II, Cairn Research, Faversham, UK) for the simultaneous observation of two spectral regions of the emitted light (EGFP: 540/75 nm (Omega Optics Filters, Brattleboro, VT, USA) and EGF-Alexa647: 675/67 nm Brightline HC (Semrock Optical Filters (IDEX Health & Science, Bristol, CT, USA), West Henrietta, NY, US), respectively). The software Andor IQ2 was used for image acquisition. The exposure time for single images was set between 300–500 ms, while the frame rate was set to the minimum. The laser power and number of focal planes (z-frames) were set for every measured cell individually between 1 and 3. Data were post-processed using ImageJ [68] (version: 2.0.0-rc-43/1.50e).

### 2.11. Image Analysis

The image analysis software “Imaris” (Bitplane, Zurich, Switzerland) was used for quantitative comparison of EGF-Alexa647 uptake over time. Z-Stacks of cells were acquired, as described before, applying the same microscopy hardware settings to ensure reproducibility between datasets and individual cells were extracted to single stacks manually. The EGF volumes were identified and rendered utilizing the “Imaris surface” feature. A surface smoothing factor of 0.05 µm was used as well as a volume thresholding based on absolute EGF signal intensity of 14. Touching objects were separated on a seed diameter of 1 µm. A set of minimum requirement feature filters were applied and repeatedly checked for quality by comparing it to non-rendered data to minimize background volumes. Final filter sets were set as following: Quality threshold of 1.89, Minimum number of Voxel of 1, Shortest axis of minimum 520 nm and minimum mean intensity of 21. The surface generation was applied in batch mode to all individual cells and mean intensity, surface volume, and surface number per cell were extracted for downstream analysis.

### 2.12. Bio-Layer Interferometry (BLI)

BLI was used to determine the binding affinity of GABARAP and EGFR derived peptides. Experiments were performed on an Octet Red 96 (FORTÉBIO, San Jose, CA, USA) while using solid-black 96-well plates at 25 °C and a shake speed of 1000 rpm. The BLI buffer contained 25 mM Tris-HCl, pH 7.5, 150 mM NaCl, 0.5 mM Tris(2-carboxyethyl)phosphin (TCEP), 0.1% Tween-20, 1 mg/mL bovine serum albumin (BSA). The EGFR peptides were synthesized and N-terminally biotinylated via an aminohexanoic acid linker as well as C-terminally amidated (CASLO, Lyngby, Denmark). GABARAP was expressed and purified, as described [69]. Peptides (50 µg/mL) or biotin (10 µg/mL) as reference were immobilized on High Precision Streptavidin (SAX) biosensors (FORTÉBIO, San Jose, CA, USA). The peptide immobilization levels were around 0.8 nm. GABARAP was used as analyte in increasing concentrations in 200 µL solution. Association of GABARAP was recorded for 300 s on ligand and reference biosensors, followed by a dissociation phase of 300 s. Steady-state evaluation was performed by plotting the respective response levels against the applied peptide concentrations. The curves were fitted according to the following Langmuir’s 1:1 binding model using OriginPro 2019 (OriginLab, Northampton, MA, USA):(1)$$y=\frac{R_{\max}x}{K_{D}+x}\;,$$
with y corresponding to the binding signal (response), R_max_ the saturation binding signal, x the applied GABARAP concentration, and K_D_ the equilibrium dissociation constant.

### 2.13. Co-immunoprecipitation

For co-immunoprecipitation (Co-IP) analysis, the GFP-Trap technology employing high affinity VHH domain containing nanobodies for GFP binding (gtak-20, Chromotek, Islandia, NY, USA) was used according to the manufacturer’s instructions. In brief, Huh7.5 GABARAP SKO cells were transfected via lipofection, as described and stimulated with 40 ng/mL EGF for 10 min. or left untreated. Afterwards, the cells were placed on ice, washed once with ice-cold PBS, and two 6-wells per condition scraped into 200 µL NP-40 lysis buffer (10 mM Tris/HCl pH 7.5, 150 mM NaCl, 0.5 mM EDTA, 0.5% Nonidet™ P40 Substitute, 0.09% sodium azide) containing protease and phosphatase inhibitors (78430, ThermoFisher Scientific). Cell lysis was carried out on ice for 30 min., with 10 s vortexing after every 10 min. Lysates were then cleared by centrifugation for 10 min. at 17 000 g at 4 °C and the supernatants containing proteins transferred to pre-cooled reaction tubes. The lysates were diluted with 300 µL ice-cold washing buffer (10 mM Tris/HCl pH 7.5, 150 mM NaCl, 0.5 mM EDTA.

0.018% sodium azide) containing protease and phosphatase inhibitors and 10% saved for input analysis. 25 µL of anti-GFP nanobody coupled agarose bead slurry was equilibrated with washing buffer, separated via a magnetic rack, and for Co-IP lysates were bound overnight at 4 °C with end-over-end mixing. Afterwards, 50 µL were saved for flow through analysis, beads were washed four times with ice-cold washing buffer, and finally eluted with 80 µL 2× Lämmli’s SDS sample buffer (120 mM Tris/HCl pH 6.8, 20% glycerol, 4% SDS, 0.04% bromophenol blue, 8% 2-mercaptoethanol) by heating to 95 °C for 5 min. The samples were then analyzed by immunoblotting, as described.

### 2.14. Statistical Analysis

All of the statistical analyses were performed with the statistical analysis software package (SPSS, version 22; SPSS Inc., Chicago, IL, USA), unless stated otherwise. Descriptive data are expressed as means ± standard error of means (SEM). Normal distribution was assessed using the Shapiro–Wilk test, and data was transformed or bias-corrected accelerated bootstrapping performed prior to analysis where necessary. Statistical testing was performed, as described individually. In general, statistical significance was inferred at a two-tailed *p*-value of ≤ 0.05. To test the influence of each GABARAP family protein on total EGFR protein levels, dichotomous dummy-coded variables were used to express each genotype as either wildtype for a specific paralog (1) or as a knockout (0). Afterwards, bivariate correlation analysis (Spearman) was performed and two-tailed statistical significance was calculated.

## 3. Results

### 3.1. Deficiency for GABARAP, but Not GABARAPL1 or GABARAPL2 Is Associated with Increased Degradation of EGFR in Huh7.5 and HEK293 Cells

We generated HEK293 knockout (KO) cells deficient for each GABARAP subfamily member alone (SKO) or in double (DKO) and triple (TKO) combination using the CRISPR/Cas9 system to systematically investigate the role of GABARAP-subfamily proteins during EGFR degradation (Appendix A).

The cells were then stimulated with 40 ng/mL EGF, which is known to promote receptor degradation [70], for up to 180 min., and whole cell lysates were analyzed for total EGFR protein levels by immunoblot (Figure 1A). Figure 1B shows densitometric analyses of the genome-edited HEK293 cell lysates compared to their matching controls. Evidently, GABARAP SKO cells displayed significantly lower EGFR levels in whole cell lysates after 10 (1.24-fold, *p* ≤ 0.05), 120 (1.84-fold, *p* ≤ 0.05), and by trend 180 min. (1.42-fold, *p* = 0.07) of EGF treatment as compared to the control levels. In contrast, neither single lack of GABARAPL1 nor GABARAPL2 led to significant differences in the total EGFR levels when compared to their respective controls, although GABARAPL1 deficiency resulted in a slight increase in EGFR at most time points, and a trend towards reduced EGFR levels could be observed for GABARAPL2 SKO cells after 180 min. of EGF treatment (1.19-fold, *p* = 0.1). Consistently, GABARAP/L1 DKO cells showed no differences in the EGFR levels as compared to the wildtype controls, neither unstimulated nor in response to EGF treatment, indicating that GABARAP and GABARAPL1 have opposite functions regarding EGFR degradation dynamics in this cell type. GABARAP/L2 DKO cells had significantly less EGFR after 10 (1.52-fold, *p* ≤ 0.05) and by trend after 60 (1.69-fold, *p* = 0.06), 120 (1.73-fold, *p* = 0.08), and 180 min. (1.9-fold, *p* = 0.1) of EGF treatment, respectively, and thus performed like lacking solely GABARAP. A lack of both GABARAPL1 and GABARAPL2 resulted in EGFR levels as in the control samples, suggesting that GABARAP is the decisive factor for a correct EGFR degradation phenotype in HEK293 cells. Notably, TKO cells, lacking the whole GABARAP subfamily, showed slightly elevated unstimulated EGFR without reaching statistical significance. This phenotype might reflect the participation of the whole GABARAP subfamily in general cellular processes such as autophagy or lysosome biogenesis [71], which, due to functional redundancy emerges, most if none of the family members are present. We used dichotomous dummy-coded variables to indicate the genotypic status for each GABARAP-subfamily member in all applied HEK293 cell lines to analyze the correlation between the presence of each of the GABARAP subfamily proteins and EGFR levels after EGF stimulation (Figure 1C) (for details see Materials and Methods).

As expected, the GABARAP availability was significantly and positively correlated with EGFR protein levels at each time point after EGF treatment (10 min., r = 0.37, *p* ≤ 0.05; 30 min., r = 0.47, *p* ≤ 0.01; 60 min., *r* = 0.41, *p* ≤ 0.01; 120 min., *r* = 0.44, *p* ≤ 0.05; 180 min., r = 0.39, *p* ≤ 0.05) as well as with the corresponding integrated area under the curve (AUC, a.u.) of EGFR levels over time (r = 0.46, *p* ≤ 0.01). 

In addition to the above described experiments, we also applied the transient overexpression of GFP-fused EGFR to the same panel of genome-edited HEK293 cell lines and analyzed their respective EGFR-(GFP) levels upon EGF stimulation by immunoblot (Appendix A). Densitometric analysis (Appendix A) largely confirmed our results that were obtained for endogenous EGFR levels. Again, GABARAP deficiency alone or in combination with GABARAPL2 deficiency resulted in a significant reduction of EGFR levels. Interestingly, GABARAPL1/L2 DKO and GABARAP/L1/L2 TKO cells both showed significantly elevated EGFR levels under EGFR overexpression conditions, suggesting that some effects observed for overexpression are either dependent on receptor density or produced by the overexpression *per se*. In general, when comparing endogenous with overexpressed EGFR degradation dynamics, it is evident that EGFR overexpression strongly slows down degradation (Appendix A, see EGFR-GFP vs. EGFR). 

Overall, GABARAP deficiency appeared to accelerate EGFR degradation or, *vice versa*, GABARAP appeared to slow down EGFR degradation upon EGF stimulation in HEK293 cells.

We next used Huh7.5 cells for degradation analysis of endogenous EGFR to clarify whether the validity of the observed GABARAP-mediated effects on EGFR degradation can be extended to other cell lines and to rule out clonal effects of our genome-edited HEK293 cell lines. To that end, we first established Huh7.5 KO cells deficient for GABARAP, GABARAPL1, or GABARAPL2 alone or combinations of GABARAP/L2 and GABARAPL1/GABARAPL2 (Appendix A). Cells and lysates were treated, as described above, total EGFR was detected by immunoblotting (Figure 2A), and the respective densitometric analyses are summarized in Figure 2B. While the basal EGFR levels were unaltered in GABARAP SKO cells, the total EGFR levels were significantly reduced after 10 (1.3-fold, *p* ≤ 0.05), 30 (2.14-fold, *p* ≤ 0.01), 60 (2.7-fold, *p* ≤ 0.05), 120 (3.56-fold, *p* ≤ 0.01), and 180 min. (4.74-fold, *p* ≤ 0.001) of treatment with EGF when compared to Huh7.5 control cells. GABARAPL1 and GABARAPL2 SKO cells were analyzed likewise with regards to EGFR protein levels in response to EGF treatment to analyze the role of the two other GABARAP subfamily members. Neither GABARAPL1 SKO nor GABARAPL2 SKO cells showed significantly reduced levels of EGFR in response to EGF stimulation when compared to Huh7.5 control cells. In fact, GABARAPL1 SKO cells displayed a slight trend towards higher EGFR total protein levels over time, while a slight but significant (1.32-fold, *p* ≤ 0.05) increase in the basal total EGFR levels was observed for GABARAPL2 SKO cells. Strikingly, the only other analyzed Huh7.5 cell line showing the accelerated degradation of EGFR in response to EGF stimulation was the GABARAP/L2 DKO line after 10 (1.9-fold, *p* ≤ 0.05), 30 (2.16-fold, *p* ≤ 0.05), 60 (2-fold, *p* ≤ 0.05), 120 (1.44-fold, *p* ≤ 0.01), and 180 min. (3.29-fold, *p* ≤ 0.05) of treatment with EGF, but not at basal levels as compared to Huh7.5 control cells. GABARAPL1/GABARAPL2 DKO cells displayed unaltered total EGFR levels when compared to Huh7.5 control cells, indicating that the lack of GABARAP alone was sufficient for driving accelerated EGFR degradation in these cells.

The correlation analysis of EGFR degradation results from Huh7.5 cell derived lysates (Figure 2C) showed broad consistency with that based on the independent HEK293 cell lysates (Figure 1C): a strong and significant positive correlation of genotypes expressing GABARAP with total EGFR levels under basal conditions (*r* = 0.4, *p* ≤ 0.05), as well as after 10 (*r* = 0.54, *p* ≤ 0.01), 30 (*r* = 0.57, *p* ≤ 0.01), 60 (*r* = 0.56, *p* ≤ 0.01), 120 (*r* = 0.66, *p* ≤ 0.001), and 180 (*r* = 0.68, *p* ≤ 0.01) min. of EGF treatment, as well as with the integrated AUC of EGFR total protein levels (*r* = 0.59, *p* ≤ 0.01) was revealed. No significant correlation for GABARAPL1 or GABARAPL2 with EGFR protein levels was found for any of the time points analyzed, although GABARAPL1 showed a trend towards negative correlation with EGFR levels at basal conditions (*r* = −0.27) and after 120 min. (*r* = −0.17) of EGF treatment. Clonal off-target effects as an explanation for the observed phenotype could be excluded with high confidence based on the close agreement of observations with two different cell lines.

In summary, the lack of GABARAP, either alone or in combination with GABARAPL2, was consistently and significantly associated with decreased total EGFR protein levels in response to EGF treatment in both Huh7.5 and HEK293 cells, whereas the presence of GABARAP in general was associated with higher EGFR total levels. Subsequently, we concentrated further efforts on the Huh7.5 GABARAP SKO cell line to analyze EGFR degradation dynamics and its implications in more detail.

### 3.2. GABARAP Deficiency Alters EGFR Signaling on Protein and Gene Expression Levels and Increases GABARAPL1 but Not GABARAPL2 Protein Expression

Next, we asked whether GABARAP deficiency is additionally accompanied by altered EGFR downstream signaling. Therefore, we analyzed the activation of EGFR itself as well as of EGFR-associated signaling, namely the phosphoinositide-3-kinase/proteinkinase B (PI3K/AKT) and mitogen-activated protein kinase/extracellular-signal regulated kinase (MAPK/ERK) signaling pathway, by immunoblot (Figure 3A). The phosphorylation of EGFR at Y1068 in GABARAP SKO cells was higher by trend after 10 min. of EGF treatment (1.36-fold, *p* = 0.06) as compared to Huh7.5 control cells, whereas it was not altered at later time points (Figure 3B). The activation of the PI3K/AKT pathway was analyzed by phosphorylation of AKT at S473 and it was not significantly influenced under the given circumstances (Figure 3C). Activating phosphorylation of ERK1/2 was reduced in GABARAP SKO cells at every time point analyzed with a significant reduction after 30 min. of EGF treatment (2.49-fold, *p* ≤ 0.05, Figure 3D).

Next, the impact of GABARAP deficiency on EGFR target gene expression was analyzed. The mRNA levels of the C-X-C chemokine receptor 2 (CXCR2) ligand *CXCL8* alongside mRNA levels of another CXCR2 ligand, *CXCL1*, were analyzed after treatment with various concentrations of EGF because the gene expression of *CXCL8*was reported to be regulated by the EGFR/ERK signaling axis after HCV infection [72]. Strikingly, *CXCL8* transcripts were significantly reduced in GABARAP SKO cells at basal levels (2.75-fold, *p* ≤ 0.05) and upon stimulation with 1.25 ng/mL (2.72-fold, *p* ≤ 0.05), 10 ng/mL (2.75-fold, *p* ≤ 0.05) and by trend 40 ng/mL EGF (3.42-fold, *p* = 0.068) for 180 min. compared to Huh7.5 control cells (Figure 3E). This was not the case for *CXCL1* (Figure 3F), suggesting different transcriptional regulation for these two chemokines through independent axes of EGFR transduced signaling.

In order to investigate the influence of GABARAP deficiency on protein levels of its paralogs GABARAPL1 and GABARAPL2, we analyzed their basal levels in Huh7.5 control and GABARAP SKO cells (Figure 3G). Interestingly, GABARAPL1 protein expression was significantly increased in GABARAP SKO cells (1.77-fold, *p* ≤ 0.05), whereas the GABARAPL2 protein levels were not consistently influenced (Figure 3H) as compared to Huh7.5 control cells.

### 3.3. Basal EGFR Surface Expression Is Unaltered in GABARAP Deficient Cells, While EGF Uptake Capacity Is Impaired over Time

We isolated the surface proteins of Huh7.5 and GABARAP SKO cells via biotinylation (Figure 4A) and determined surface EGFR levels by immunoblot (Figure 4B) in order to check whether accelerated degradation of EGFR is simply caused by altered EGFR surface expression. Densitometric analysis revealed no alterations between GABARAP SKO and Huh7.5 control cells (Figure 4C), indicating that the trafficking of EGFR to the plasma membrane is not impaired in GABARAP SKO cells under basal levels of EGF. Consistent with unaltered EGFR total protein levels, the *EGFR* mRNA expression levels were not influenced by GABARAP deficiency (Figure 4D). We next asked whether the observed acceleration of EGFR degradation and alterations in downstream signaling events are caused by defects in EGF uptake or receptor endocytosis *per se*. Therefore, we carried out a FACS based assay. The cells were continuously treated with 40 ng/mL EGF that was conjugated to the fluorophore Alexa647 and the median fluorescence intensity (a.u.), reflecting the amount of intracellular EGF, was analyzed via FACS (Figure 4E). As EGFR is the only receptor for EGF, this directly reflects its internalization by endocytosis at early time points and accumulated intracellular EGF over time. However, Alexa647 is a pH-stable fluorophore and it is therefore not fully degraded by lysosomes similar to EGF quantum dots [73]. Hence, intracellular fluorescence at later time points might reflect free dye, at least to some extent. Nonetheless, this approach represented several cycles of EGFR internalization, which provides a measure of EGF uptake over time. Consistent with surface expression being unaltered, GABARAP SKO cells displayed unaltered median fluorescent intensity values after 10 min. of EGF-Alexa647 treatment, indicating that early internalization events are not impaired in GABARAP deficient cells. Over time, intracellular EGF-Alexa647 levels were lower in GABARAP SKO cells after 30, 60, and 120 min. by trend and significantly reduced after 180 min. (1.59-fold, *p* ≤ 0.01) when compared to Huh7.5 control cells (Figure 4F). The obtained results with this continuous treatment conditions, which allow for several rounds of ligand binding and receptor cycling, indicate a shift from receptor recycling to degradation in GABARAP SKO cells at later time points as compared to Huh7.5 control cells. This is in line with the immunoblot experiments and suggests that the reduction of EGF-Alexa647 levels is caused by a general reduction of EGFR protein levels over time in response to EGF treatment in GABARAP deficient cells.

### 3.4. Tracking of Fluorescently-Labeled EGF Reveals Altered EGF Trafficking and Vesicular Composition in GABARAP Deficient Cells without Abrogation of General Endosomal Targeting

We used a pulse-based setup for confocal laser scanning microscopy imaging to address the question of whether GABARAP deficiency causes impaired intracellular trafficking of EGFR. The cells were pre-incubated with 40 ng/mL EGF-Alexa647 at 4 °C to saturate all EGFR binding sites at the plasma membrane, followed by acidic wash to remove unbound EGF and subsequent incubation at 37 °C to allow for internalization and trafficking of EGF pre-bound to EGFR (Figure 5A). As exemplarily shown for 5, 30 60, and 120 min. (Figure 5B), the cells were fixed at various time points after the EGF pulse and EGF accumulation was recorded as distinct dots, likely reflecting that EGFR accumulated in vesicles subsequent to ligand stimulation. EGF-Alexa647 containing vesicles were analyzed regarding the average number (Figure 5C), volume (Figure 5D) and intensity (Figure 5E) of individual vesicles per cell. Overall vesicular number per analyzed cell was significantly reduced after 60 (1.21-fold, *p* ≤ 0.05) and 120 min. (1.29-fold, *p* ≤ 0.05) of EGF-Alexa647 incubation. While the overall vesicular volume at 5 min. after the EGF-Alexa647 pulse was significantly increased in GABARAP SKO cells when compared to Huh7.5 control cells (1.15-fold, *p* ≤ 0.05), it was found to be significantly lower after 30 min. (1.26-fold, *p* ≤ 0.001) as compared to Huh7.5 control cells. Accordingly, the mean fluorescent intensity of analyzed vesicles after 5 min. of EGF-Alexa647 stimulation was significantly higher in GABARAP SKO cells when compared to Huh7.5 control cells (1.04-fold, *p* ≤ 0.05), while it was found to be significantly decreased after 30 min. (1.07-fold, *p* ≤ 0.001).

Next, we classified the obtained vesicle volumes and intensities into subgroups to visualize the more subtle differences that are not resolved by the global approach described above. The corresponding plots are shown in Appendix A. In summary, for vesicles of the smallest category (0.5–1 µm^3^) similar numbers were found in GABARAP SKO and Huh7.5 control cells at all time points. Interestingly, for all other size categories and particularly at later stages starting at 30 min. post EGF-Alexa647 treatment, the numbers of observed vesicles were decreased in GABARAP deficient cells. Solely for the early 5 min. time point, increased numbers of vesicles of several categories were found for GABARAP SKO cells when compared to Huh7.5 control cells. Accordingly, vesicles of the lowest intensity (20–40) were not altered between GABARAP SKO and Huh7.5 cells, while vesicles of the remaining three intensity categories were significantly decreased, again especially starting at 30 min. post EGF-Alexa647 treatment. EGF-Alexa647 pulse-treated GABARAP SKO and Huh7.5 control cells were fixed and stained for early (RAB5), recycling (RAB11) or late (RAB7) endosomal marker proteins, to further analyze endosomal trafficking. Colocalization events of RAB5, RAB11, and RAB7 with EGF-Alexa647 (white arrows) were observed for GABARAP SKO and Huh7.5 control cells at every analyzed time point, indicating that the general ability of EGFR to reach the analyzed endosomal compartments is not abolished by GABARAP deficiency, as exemplarily shown for 10, 30, and 60 min. in Appendix A. However, fixed cells and the analyzed set of time points might not be suitable to capture subtle and transient GABARAP-mediated interactions.

### 3.5. Accelerated EGFR Degradation in GABARAP Deficient Huh7.5 Cells Can Be Counteracted by Lysosomal and Proteasomal Inhibition

We then asked whether the acceleration in EGF-induced EGFR degradation in GABARAP deficient cells depends on the activity of the proteasomal or lysosomal machinery or whether degradation would occur through a different non-canonical mechanism. To this end, the inhibition of lysosomal or proteasomal activity was obtained by chloroquine or lactacystin treatment, respectively (Figure 6A), which are known to inhibit either lysosomal acidification (chloroquine) or proteasomal subunits (lactacystin) and cause delay in EGF-induced EGFR degradation [74,75,76]. The cells were treated with chloroquine and total EGFR levels as well as the activation of MAPK/ERK signaling in response to subsequent EGF stimulation was analyzed by immunoblot (Figure 6B). This led to a delay in EGF-induced EGFR degradation in Huh7.5 cells and could, at least partly, restore the declined EGFR levels observed in GABARAP SKO cells in response to EGF stimulation shown in Figure 2A, as the total EGFR levels were only significantly reduced after 60 min. (1.69-fold, *p* ≤ 0.05), but not at any of the other time points compared to Huh7.5 control cells (Figure 6C). Interestingly, MAPK signaling assessed by activating phosphorylation of ERK1/2 was still found to be significantly reduced in GABARAP SKO cells after 30 (1.77-fold, *p* ≤ 0.01), 120 (1.8-fold, *p* ≤ 0.05), and 180 min. (1.82-fold, *p* ≤ 0.01) of EGF treatment when compared to controls (Figure 6D). Next, lactacystin treatment was applied prior to EGF stimulation and immunoblot analysis (Figure 6E).

Interestingly, this led to a delay in EGF-induced EGFR degradation in Huh7.5 control cells and abrogated the differences in EGFR protein between GABARAP SKO and Huh7.5 control cells after stimulation, as shown in Figure 2A, at any of the analyzed time points (Figure 6E). MAPK/ERK signaling was also found to be restored after lactacystin treatment in GABARAP SKO cells when compared to Huh7.5 control cells (Figure 6F). Taken together, these results indicated that GABARAP deficiency does not change the mechanism of EGFR degradation in general, but rather affects upstream events related to receptor trafficking. Impaired ERK signaling of GABARAP SKO cells under chloroquine treatment points to endosomal trafficking events when the cytoplasmic tail of the receptor is still able to contact the cytoplasm to activate downstream signaling molecules.

### 3.6. GABARAP and EGF Converge in Distinct Dynamic Vesicular Structures at Endogenous Expression Levels in HEK293 Cells

We asked whether both molecules localize to the same endosomal compartment after ligand stimulation to obtain an insight into the trafficking events underlying GABARAP-mediated regulation of EGFR degradation. We generated a knock-in (KI) cell line expressing GFP-tagged GABARAP under control of the endogenous *GABARAP* promoter by CRISPR/Cas9 mediated genome editing to eliminate the impact of overexpression artifacts (Appendix A). The resulting GFP-GABARAP expression levels were sufficient for live cell microscopy, as demonstrated in Figure 7. 

GABARAP was found to be present in distinct structures in the cell’s cytoplasm. These structures displayed different characteristics regarding their shape, size, and cargo, as determined by the simultaneous use of EGF-Alexa647 and Transferrin (Tf)-Alexa555 (Figure 7A, Appendix A). The latter was applied as a marker for endosomal compartments associated with recycling [77]. GABARAP and EGF frequently converged in punctate structures (yellow arrows), indicating that both of the molecules are located within the same vesicle or at least adjacent vesicular structures. Some of these vesicles were additionally observed to be Tf-Alexa555 positive (white arrows), while others were found to be EGF and Tf double-positive (magenta arrows) without GABARAP localization. In contrast, we rarely observed GABARAP vesicles, which were additionally only Tf-positive. GABARAP and EGF also converged in Tf-negative ball-shaped structures (yellow arrows), indicating the accumulation of EGF and GABARAP within the same endosomal compartment, potentially associated with the degradative branch.

Strikingly, we frequently found large ring-like structures that were labeled with GABARAP (Appendix A) of up to 3 µm in diameter. They were found at most once per cell and vesicles either double-positive for EGF and Tf or single-positive for EGF fused with the perimeters of these rings or budded off them. EGF accumulation was found in clusters resembling microdomains on these rings. Some, but not all, of these EGF clusters also contained Tf, suggesting that the respective parts of such rings might be associated with recycling. Altogether, these observations suggested that the large GABARAP-positive ring-like structures represent some sort of endosomal compartment, potentially a sorting endosome at the center of endosomal targeting either towards recycling or degradation.

We then subjected the GFP-GABARAP KI line to spinning disk confocal fluorescence microscopy to increase the temporal resolution (Figure 7B–D, Appendix A). After EGF-Alexa647 treatment, we could observe highly dynamic vesicular structures that were constantly fusing with and budding off the aforementioned GABARAP-positive rings. Figure 7B illustrates the fusion of a GABARAP single-positive vesicle with such a ring within a time frame as short as 1.2 s (green arrows). We also observed GABARAP vesicles that were EGF-Alexa647 positive (yellow arrows) and budded off the rings in a coordinated manner (Figure 7C). Frequently, these budding events were preceded by tubular protrusions (Figure 7D), which might represent molecules destined for recycling. Indeed, such cargo has been described to be sorted by tubular endosomal structures [78].

### 3.7. GABARAP Associates with EGFR during Co-Immunoprecipitation and Binds to Synthetic Peptides Derived from the EGFR Cytoplasmic Tail

We performed co-immunoprecipitation experiments, followed by *in vitro* interaction studies using purified GABARAP and synthetic peptides derived from the EGFR cytoplasmic tail, to investigate the nature of the transient co-migration observed during live cell imaging. As shown in Figure 8A, EGFR was co-immunoprecipitated by GFP-GABARAP but not by GFP from lysates of transiently transfected GABARAP SKO cells. This experiment confirmed an association between GABARAP and EGFR within cells. Interestingly, association was observed both under unstimulated conditions and after 10 min. of EGF treatment, supporting the idea of a GABARAP effect early in EGFR trafficking after ligand stimulation. The observed spatial overlap between GABARAP and EGFR during our live cell imaging studies, together with their observed co-immunoprecipitation, finally encouraged us to scan the EGFR sequence for canonical LIR/GIM motifs as putative direct GABARAP-binding sites. To address this, we used the iLIR tool [79]. Interestingly, EGFR indeed includes a putative extended LIR motif (xLIR) encompassing positions 1060 to 1065 (DTFLPV) within its cytoplasmic, regulatory tail (Figure 8B). Overall, this LIR motif contains four negatively charged and three phosphorylatable residues are located between P-8 and P-1, a further negatively charged residue at P + 5 and two phosphorylatable residues at P + 6 and P + 10. These features are in line with the established LIR-motifs of well-known GABARAP interactors, as demonstrated by alignment with the respective regions of ULK1, autophagy-related protein 13 (ATG13), Sequestosome-1 (SQSTM), pericentreolar material 1 (PCM1), and FIP200 [80].

Two aminoterminally biotinylated peptides covering the xLIR “DTFLPV” and additional eight positions up- and ten positions downstream, one of them phosphorylated at the regulatory Y1068, were subjected to biolayer interferometry (BLI) to analyze the binding affinity of GABARAP to this EGFR region. Figure 8C shows results of BLI measurements. The obtained dissociation constants were 96.5 µM (±5.1 µM) and 82 µM (±3.3 µM) for the unmodified and the modified peptide, respectively. These affinities appear relatively weak when compared to those of other known GABARAP protein interactors being in the low micromolar range [49]. Nonetheless, this interaction still might be of relevance e.g., in microdomains of locally clustered EGFR and GABARAP molecules (Figure 8D) by increasing overall avidity. Whether or not the xLIR motif is decisive for GABARAP binding to EGFR will be the subject of further investigation.

## 4. Discussion

In this study we identified a unique and novel role for GABARAP in EGF-induced trafficking and degradation of the EGFR, with implications for EGFR downstream signaling. Based on two independently generated HEK293 and Huh7.5 KO cell line panels, we could show that only cells lacking GABARAP, but not GABARAPL1 or GABARAPL2, displayed reduced total EGFR protein levels after EGF stimulation. We further showed that MAPK signaling downstream of EGFR was impaired in GABARAP deficient Huh7.5 cells, which translated into the reduction of EGFR target gene *CXCL8* expression. Consequently, we then explored the potential mechanistic role of GABARAP in the context of EGFR trafficking and degradation.

EGFR cycling can roughly be divided into five stages (Figure 9A): EGFR gene transcription and protein expression (1) are followed by post-translational modifications in the ER, trafficking through the Golgi-apparatus and surface targeting (2). Plasma membrane localized EGFR can then encounter extracellular stimuli, such as EGF. Ligand-bound EGFR is activated and internalized to strictly control signaling strength and duration (3). Subsequently, EGFR gets sorted within the endosomal system and is either recycled back to the plasma membrane (4) or targeted for degradation in the lysosome (5).

An impact of GABARAP on basal protein levels of both total and cell surface localized EGFR seems unlikely, given our observations under unstimulated conditions. Additionally, steady state and EGF stimulated gene expression of the receptor was not influenced by GABARAP deficiency.

While we also did not observe an impact of GABARAP on the initial uptake of EGF-Alexa647, several hints strongly suggest that early internalization events might be slowed down by GABARAP. We found GABARAP deficiency associated with initially increased phosphorylation of EGFR Y1068, which is known to be associated with EGFR activation, growth factor receptor-bound protein 2 (GRB2), and E3 ubiquitin-protein ligase CBL binding, followed by internalization and subsequent degradation of the receptor [81]. We also observed higher volumes and intensities of EGF-containing vesicles as early as after 5 min. of EGF treatment in GABARAP deficient cells, indicating that GABARAP negatively influences the speed of early uptake events. Taken together, these observations suggest that GABARAP acts at an early stage of endosomal EGFR trafficking immediately downstream of ligand-induced receptor activation. In that way, increased EGFR degradation in GABARAP SKO cells would be a cumulative effect based on influencing early receptor dynamics.

Whether GABARAP influences EGFR activation through regulating the strength of dimer formation, as reported for different EGFR ligands [82], needs to be determined in further studies. As exemplarily shown in Figure 9B, a plethora of GABARAP interaction partners have already been reported to participate in endosomal sorting of the EGFR. The internalization of EGFR via clathrin-mediated endocytosis (CME) [16] might be modulated through direct interaction of GABARAP with the clathrin-heavy chain, which has already been described [83]. Interestingly, *CXCL8* expression was already reduced at low (i.e., 1.25 ng/mL) concentrations of EGF in the absence of GABARAP, indicating that CME, which is the major internalization route at low ligand concentrations [17], might be positively influenced by GABARAP. The high ligand concentrations that we mainly used in our study are known to activate clathrin-independent endocytosis (CIE), balancing the ratio of CME:CIE to about 1:1 [70]. CIE was reported to be mediated by ER/plasma membrane contact sites facilitated by reticulon 3 (RTN3) [84]. Intriguingly, RTN3 was recently described to interact with human Atg8 proteins through functional LIR motifs in the context of selective ER-phagy [85]. Thus, GABARAP could also sequester RTN3, hindering it from promoting CIE of EGFR. Thereby, GABARAP might shift the equilibrium towards enhanced CME and recycling.

Additionally, the E3 ubiquitin ligases NEDD4 and CBL have already been described to exhibit functional LIR motifs [25,86]. They take part in monoubiquitination of activated receptors, which is a signal for sorting into degradative compartments [87,88]. In particular, NEDD4 facilitates EGF-induced EGFR degradation by the ubiquitination of activated Cdc42-associated tyrosine kinase (ACK), leading to the degradation of both proteins [89]. Members of the E3 ubiquitin-protein ligase CBL family also target receptor tyrosine kinases for degradation by ubiquitination [90]. GABARAP might sequester these E3 ligases and, thus, prevent them from targeting receptors to degradation by monoubiquitination. With Cullin-3 (CUL3) another E3 ligase was reported to be positively involved in the maturation of late endosomes [91] and interact with GABARAP via KBTBD6/7 (Kelch repeat and BTB domain-containing protein 6/7) binding [92]. In this context, GABARAP might indirectly sequester CUL3 via KBTBD6/7, thereby inhibiting its positive effect on late endosome maturation and thus attenuate EGFR degradation.

Experimental evidence for GABARAP participating at the level of endosomal sorting comes from our EGF uptake results using confocal laser scanning microscopy, demonstrating that GABARAP SKO cells show altered vesicular size and EGF-Alexa647 loading, especially regarding larger vesicles with high fluorescence intensity at later time points upon stimulation.

The RAS-related in brain (RAB) protein family of small GTPases plays a major role in endocytic trafficking [93]. Several possibilities exist for GABARAP to influence RAB related processes. TBC1 domain family member 16 (TBC1D16) was described to be a negative regulator of RAB4A, thereby inhibiting the recycling of activated EGFR [94]. TBC1D16 was also reported to interact with Atg8 proteins in pulldown experiments, similar to other TBC domain containing proteins [95]. Whether GABARAP directly interacts with TBC1D16 to counteract its negative effect on EGFR recycling or whether other TBC domain containing proteins are involved needs to be elucidated in future studies.

The association of the RAB proteins 5, 11, and 7 with EGF-Alexa647 was not found to be altered in GABARAP deficient cells during our experiments, implicating that GABARAP activity is not necessary for general endosomal targeting of EGFR. However, we cannot exclude that subtle or transient differences remained undetected under the experimental conditions used. Time-lapse live cell imaging while using KI cell lines expressing fluorescent protein tagged RABs can help to clarify this issue in more depth in the future.

KI cells expressing GFP-GABARAP under the endogenous *GABARAP* promoter enabled us to detect transient co-migration of GABARAP and EGF in the cytoplasm of cells under live conditions while using confocal laser scanning microscopy. Diverse punctate vesicular structures were frequently found along with GABARAP positive rings forming microdomain-like spots, which were either positive for EGF- or for the recycling compartment marker Transferrin, emphasizing the importance of GABARAP-related activity for EGFR trafficking. Using spinning disk confocal fluorescence microscopy we improved the temporal resolution up to 20-fold, enabling us to assess the dynamics of GABARAP- and EGF-containing vesicles. We observed highly dynamic GABARAP-, EGF- or GABARAP/EGF-containing vesicles fusing with or budding off such rings. These intracellular interactions strongly suggest that also later stages of endosomal trafficking that are not associated with Tf are affected by GABARAP, potentially correlating with a role in endosomal sorting and/or maturation.

A potential role of GABARAP in inhibiting endosome maturation is supported by the finding that the protein levels of the CCZ1/MON1 positive regulator RMC1 are elevated in cells deficient for the whole GABARAP subfamily [41]. CCZ1/MON1 acts as an activator of RAB7 [96]. The inhibition of late endosome maturation might be mediated by GABARAP preventing RMC1 from activating RAB7 through CCZ1/MON1. Another RAB regulator interacting with GABARAP is the RAC1 GEF Ost-III, which negatively regulates CME of receptors and it was found to be inhibited by ectopic GABARAP expression [97]. Two main degradative pathways play a role in EGF-induced EGFR degradation. First, proteasome-mediated deubiquitination of activated receptors is necessary for EGFR containing endosomes to mature into intraluminal vesicles (ILV) of multivesicular bodies (MVB) [98]. Second, processed receptors are targeted for degradation within the lysosomal compartment [99]. Proteasomal inhibition by lactacystin restored both EGFR protein levels and ERK1/2 phosphorylation to wildtype, indicating that GABARAP acts downstream or on the level of MVB maturation. In contrast, the inhibition of lysosomal acidification by chloroquine partly restored EGFR degradation towards wildtype levels. Notably, ERK1/2 phosphorylation was still impaired in GABARAP SKO cells. Chloroquine is known to inhibit EGFR degradation by preventing fusion of multivesicular bodies/late endosomes with the lysosome [100], indicating that GABARAP affects ERK1/2 activation earlier in the process. Thus, we hypothesize that GABARAP might act on the level of endosomal maturation and/or compartmentalization, e.g., by controlling maturation of EGFR containing vesicles into ILVs of MVBs upstream of lysosomal degradation.

Finally, Pleckstrin homology domain-containing family M member 1 (PLEKHM1) binding simultaneously to Atg8 family proteins and the homotypic fusion and protein sorting (HOPS) complex was reported to regulate ligand-induced EGFR degradation due to impaired lysosomal fusion [101]. Importantly, PLEKHM1 was found to contain a LIR displaying a much higher affinity to GABARAP subfamily proteins than LC3 subfamily proteins [49]. However, the binding of all GABARAP subfamily proteins was described to be in the low micromolar range, strongly suggesting that GABARAP function in this context might be redundant to GABARAPL1 and/or GABARAPL2.

GABARAP might also directly bind to EGFR under certain circumstances due to the existence of an xLIR motif, which we identified in the cytoplasmic domain of EGFR. The binding of EGFR LIR peptides to immobilized GABARAP was quite modest during our measurements as compared with most GABARAP interactions reported previously. Nevertheless, EGF stimulation is known to promote EGFR nanocluster formation [102,103] by receptor oligomerization and membrane bending [104] prior to internalization [105]. GABARAP has also been described to form self-associated species [106]. Such clusters may increase the local EGFR and/or GABARAP concentration facilitating LIR-mediated binding. Indeed, we were able to show the co-immunoprecipitation of GFP-GABARAP and EGFR both with and without EGF treatment. These results and the observed comigration of GFP-GABARAP and EGF-Alexa647 strongly suggest an at least transient interaction between GABARAP and EGFR.

Such an interaction could either result in direct targeting of EGFR into autophagosomes or involve endosomal sorting. Direct autophagic targeting of proteins by GABARAP has recently been described for the nuclear receptor co-repressor 1 (NCOR1) [107]. On the other hand, EGFR activation actively suppresses autophagy by beclin 1 phosphorylation [108], and we did not use autophagy inducing conditions in our set up.

GABARAP might bind to EGFR-containing vesicles, presenting the xLIR motif on their outer face and thereby connect them to the microtubule network, which is known to associate with GABARAP [52]. In this case, EGFR vesicle transport would be mediated in a very direct manner.

Finally, several RTKs are associated with autophagy related processes. For example, protein turnover of TNFRSF12A (TNF receptor superfamily member 12 A) is regulated by mammalian Atg8 family proteins, with GABARAP and GABARAPL2 fulfilling different roles in this process [109]. Autophagy degraded the RTK ret proto-oncogene (RET) [24]. The former case supports the idea of non-redundant roles of GABARAP subfamily proteins, similar to what we observe for EGFR in Huh7.5 cells regarding GABARAP action.

## 5. Conclusions

Altogether, the presented data supports a unique and non-redundant role for GABARAP in the context of EGF-induced EGFR degradation. GABARAP may be able to influence EGFR trafficking on numerous levels, including, but very likely not limited to, a direct interaction with EGFR, as depicted in Figure 9. Therefore, further detailed studies will be necessary to determine the underlying molecular mechanism(s) of GABARAP interfering with EGFR trafficking and endosomal trafficking in general. It will also be of paramount importance to clarify the roles of the other two GABARAP subfamily proteins in that context. Lastly, we shall not forget the involvement of GABARAP subfamily proteins in important cellular processes, such as autophagy and lysosomal fusion, which cannot be ruled out to have an effect on most phenotypes in general. We have just started to uncover the mode of action of human Atg8 proteins and their contribution to cell surface receptor fate in general.

## Figures and Tables

**Figure 1 cells-09-01296-f001:**
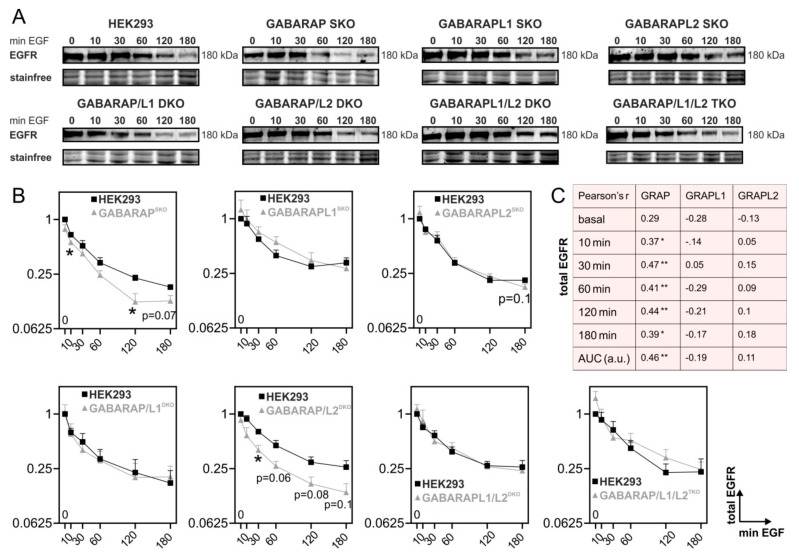
Epidermal growth factor (EGF)-induced EGF receptor (EGFR) degradation in HEK293 knockout (KO) cells. (**A**) Cells were treated with 40 ng/mL EGF for the indicated times. Afterwards, total EGFR protein levels in whole cell lysates were analyzed by immunoblot. Representative blots are shown for at least n = 3 independent experiments. (**B**) Densitometric analysis of at least n = 3 independent experiments. Controls are plotted for each experiment. Quantification of total EGFR protein levels was performed by normalization to stain-free loading control; levels are given relative to HEK293 control cells at unstimulated conditions (t = 0). Error bars represent standard error of means. Asterisks mark significant differences versus the corresponding time point of control cells as calculated using independent t-test. *p* ≤ 0.05 = *, *p* ≤ 0.01 = **, *p* ≤ 0.001 = ***. (**C**) Correlation of presence of each GABARAP with total EGFR levels in response to EGF treatment. Correlations were calculated taking every analyzed KO combination except GABARAP/L1/L2 TKO in HEK293 cells into account. Pearson correlation was used for calculation and two-tailed significances are denoted with asterisks: *p* ≤ 0.05 = *, *p* ≤ 0.01 = ** GRAP = GABARAP, GRAPL1 = GABARAPL1, GRAPL2 = GABARAPL2. Respective wildtype controls were run on the same PAGE for each KO cell line and can be found in Appendix A which also shows the uncropped source blots.

**Figure 2 cells-09-01296-f002:**
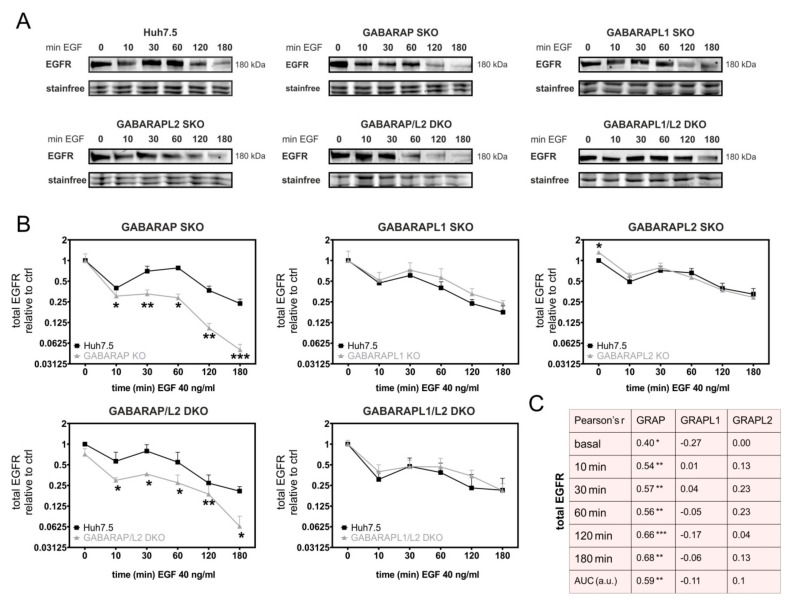
EGF-induced EGFR degradation in Huh7.5 KO cells. (**A**) Cells were treated with 40 ng/mL EGF for the indicated times. Afterwards, total EGFR protein levels in whole cell lysates were analyzed by immunoblot. Representative blots are shown for n = 3 independent experiments. (**B**) Densitometric analysis of n = 3 independent experiments. Controls are directly plotted for each experiment. Quantification of EGFR protein levels was performed by normalization on stain-free loading control and calculated as percentage of Huh7.5 control cells at unstimulated conditions (t = 0). Representative blots are shown for a summary of n = 3 independent experiments. Error bars represent standard error of means. Asterisks mark significant differences versus the corresponding time point of control cells as calculated using independent t-test. *p* < 0.05 = *, *p* < 0.01 = **, *p* < 0.001 = ***. (**C**) Correlation of presence of each GABARAP with total EGFR levels in response to EGF treatment. Correlations were calculated taking every analyzed KO combination in Huh7.5 cells into account. Pearson correlation was used for calculation and two-tailed significances are denoted with asterisks*: p* ≤ 0.05 = *, *p* ≤ 0.01 = **. GRAP = GABARAP, GRAPL1 = GABARAPL1, GRAPL2 = GABARAPL2. Respective wildtype controls were run on the same PAGE for each KO cell line and can be found in Appendix A which also shows the uncropped source blots.

**Figure 3 cells-09-01296-f003:**
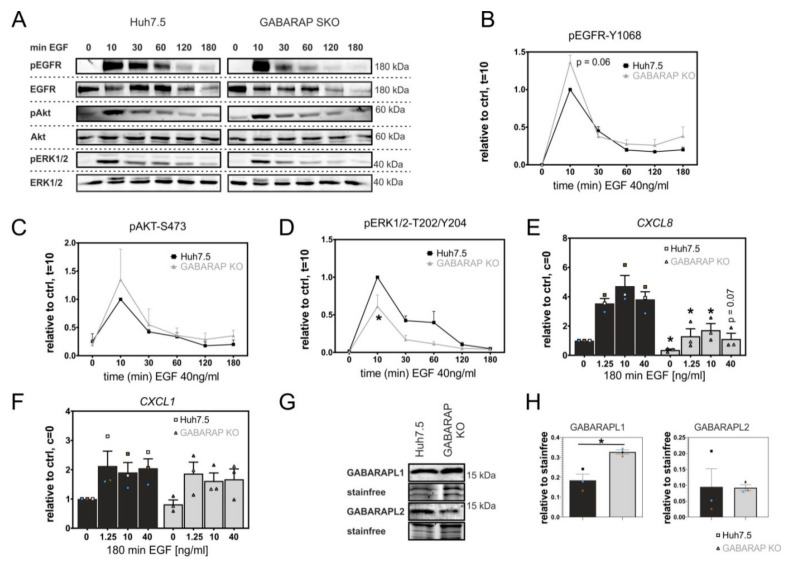
Analysis of EGF-induced EGFR phosphorylation, activation of downstream signaling and gene expression in GABARAP SKO and Huh7.5 control cells. (**A**) Huh7.5 and GABARAP SKO cells were treated with 40 ng/mL EGF for the indicated times. Afterwards, activating phosphorylations of the EGFR (**B**) and downstream PI3K/AKT (**C**) and MAPK/ERK (**D**) signaling pathways were analyzed by immunoblot. Quantification of phosphorylated proteins was performed by normalization to the corresponding total protein levels and calculated as percentage of Huh7.5 control cells after 10 min. of EGF treatment (t = 10). (**E** + **F**) Huh7.5 and GABARAP SKO cells were treated with the indicated concentrations of EGF for 180 min., followed by cell lysis, RNA extraction, reverse transcription and quantitative real-time PCR (RT-PCR). Expression of target genes *CXCL8* (**E**) and *CXCL1* (**F**) was normalized to *succinate dehydrogenase* (*SDHA*) as reference gene and is expressed relative to unstimulated control cells. (**G**) Protein expression of GABARAP paralogs GABARAPL1 and GABARAPL2 was analyzed in Huh7.5 and GABARAP SKO cells and densitometric analysis performed to determine GABARAPL1 and GABARAPL2 protein levels in GABARAP SKO cells compared to Huh7.5 controls (**H**). Representative blots are shown for a summary of n = 3 independent experiments. Error bars represent standard errors of means. Asterisks mark significant differences versus the corresponding time point or concentration of control cells as calculated using independent t-test. *p* ≤ 0.05 = *. (**E**,**F**,**H**) individual experiments are color-coded. Appendix A shows uncropped source blots.

**Figure 4 cells-09-01296-f004:**
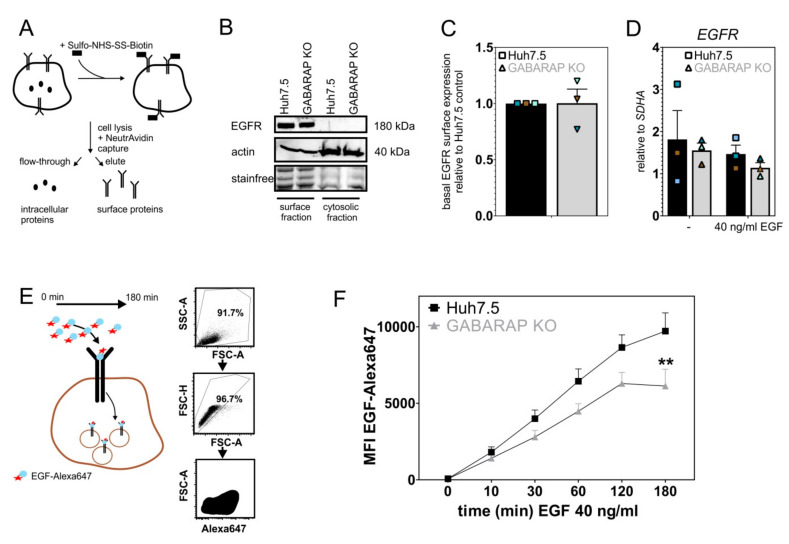
Analysis of EGFR surface expression by isolation of surface proteins and EGF-uptake in GABARAP SKO and Huh7.5 control cells by flow cytometry. (**A**) Primary amines of extracellular portions of plasma membrane proteins were conjugated to Sulfo-NHS-SS-Biotin. Afterwards, cells were lysed and biotinylated proteins captured via NeutrAvidin to separate surface from cytosolic proteins. (**B**) EGFR surface expression of Huh7.5 control and GABARAP SKO cells was determined by immunoblotting of the surface fraction lysate and (**C**) densitometric analysis performed to determine relative EGFR protein surface expression levels between Huh7.5 and GABARAP SKO cells (n = 3 independent experiments). (**D**) EGFR mRNA expression levels were analyzed at steady-state and after 180 min. of 40 ng/mL EGF treatment. Expression levels were normalized to *SDHA* expression and compared between Huh7.5 and GABARAP SKO cells. (**E**) Huh7.5 and GABARAP SKO cells were continuously treated with 40 ng/mL of Alexa647 labelled EGF over 180 min. Gates were set to get rid of debris and select for single cells. Intracellular EGF was determined by analyzing median fluorescence intensity (MFI) of EGF-Alexa647 positive cells (**F**). Line plot is a summary of n = 4 independent experiments. Error bars represent standard errors of mean. Asterisks mark significant differences to the corresponding time point of control cells as calculated using two-way analysis of variance with Bonferroni post-hoc testing with GraphPad Prism version 8.00 for Windows (GraphPad Software, La Jolla California US, www.graphpad.com). *p* ≤ 0.01 = **. (**C** + **D**) Individual experiments are color-coded. Appendix A shows uncropped source blots.

**Figure 5 cells-09-01296-f005:**
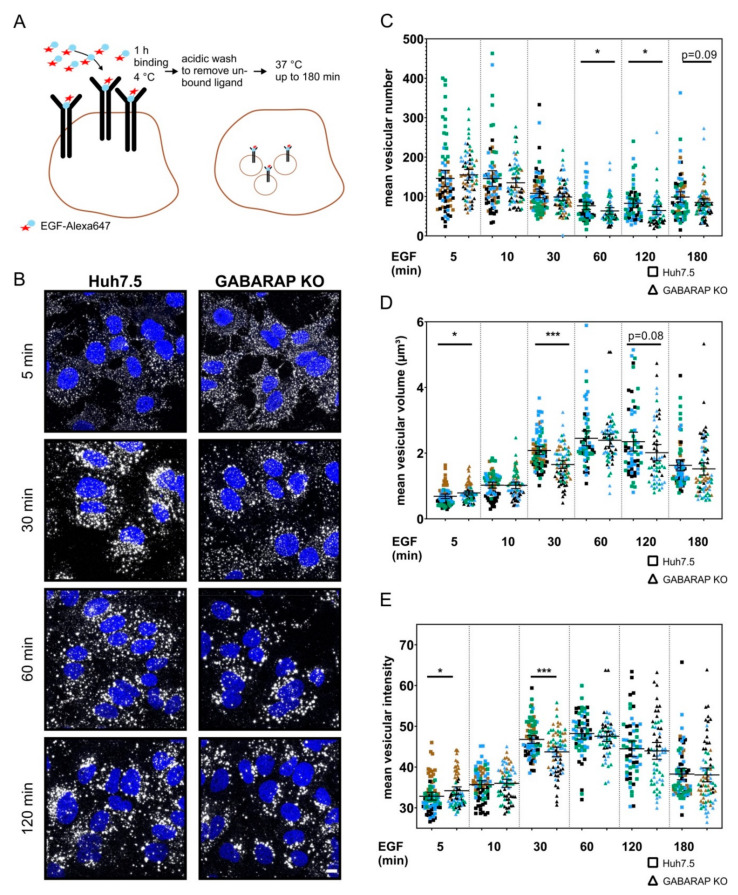
Analysis of EGF uptake in GABARAP SKO and Huh7.5 control cells via EGF-Alexa647 pulse by immunofluorescence imaging. (**A**) Huh7.5 and GABARAP SKO cells were treated with 40 ng/mL Alexa647 labelled EGF at 4 °C to allow binding to EGFR. After rigorous washing, cells were placed at 37 °C and analyzed at distinct time points to assess EGF internalization. (**B**) Cells were fixed at basal levels and after 5, 10, 30, 60, 120, and 180 min. of EGF treatment. Nuclei were counterstained with DAPI and images of EGF positive puncta taken. Mean vesicular number (**C**), volume (**D**) and intensity (**E**) of imaged EGF-Alexa647 puncta were analyzed by Imaris. Single cells were cropped out of images, processed with ImageJ and Imaris (Bitplane, Zurich, Switzerland). Only vesicles with volumes > 0.1 µm^3^) were taken into account. Fluorescent images are representative of at least n = 3 independent experiments. Individual experiments are color-coded; >50 cells per genotype and time point were analyzed. Error bars represent 95% confidence intervalls (CI) (**C**–**E**). Asterisks mark significant differences to the corresponding time point of control cells and were calculated using independent t-test*. p* ≤ 0.05 = *, *p* ≤ 0.01 = **, *p* ≤ 0.001 = ***.

**Figure 6 cells-09-01296-f006:**
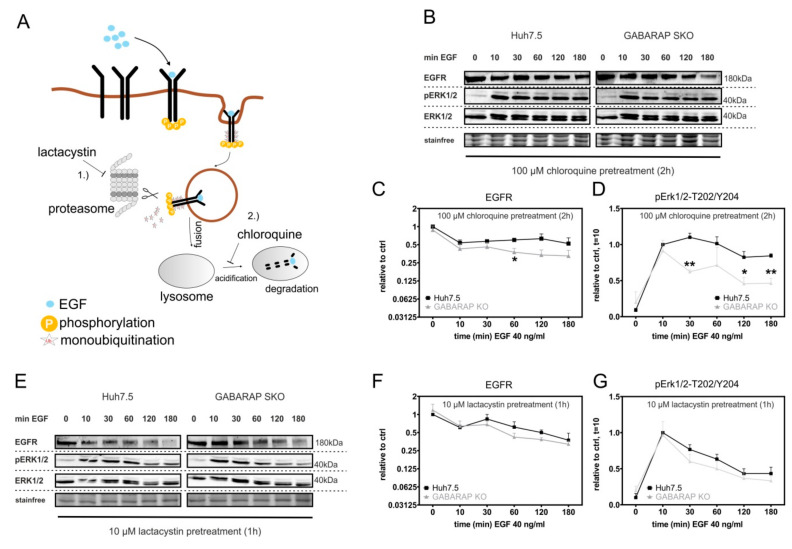
EGF-induced EGFR degradation after inhibition of lysosomal acidifcation and proteasomal inhibition in GABARAP SKO and Huh7.5 control cells. (**A**) Modes of action of the inhibitors used. Cells were pretreated with inhibitors of lysosomal acidifcation or proteasomal subunits and afterwards treated with 40 ng/mL EGF for the indicated times. (**B**–**D**) Cells were treated with lysosomal acidification inhibitor chloroquine. Total EGFR levels as well as activating phosphorylation of ERK1/2 at T202/Y204 were analyzed by immunoblotting and densitometry. (**D**–**G**) Cells were treated with proteasomal subunit inhibitor lactacystin. Total EGFR levels as well as activating phosphorylation of ERK1/2 at T202/Y204 were analyzed by immunoblotting and densitometry. Quantification of protein levels was performed by normalization to stain-free protein loading or the respective total levels of downstream signaling proteins and calculated as percentage of Huh7.5 control cells at unstimulated conditions (t = 0) for EGFR and peak activation levels (t = 10 min.) for ERK. Representative blots are shown for a summary of n ≥ 3 independent experiments. Error bars represent standard errors of means. Asterisk marks significant difference versus the corresponding time point of control cells as calculated using independent t-test. *p* ≤ 0.05 = *. Appendix A shows uncropped source blots.

**Figure 7 cells-09-01296-f007:**
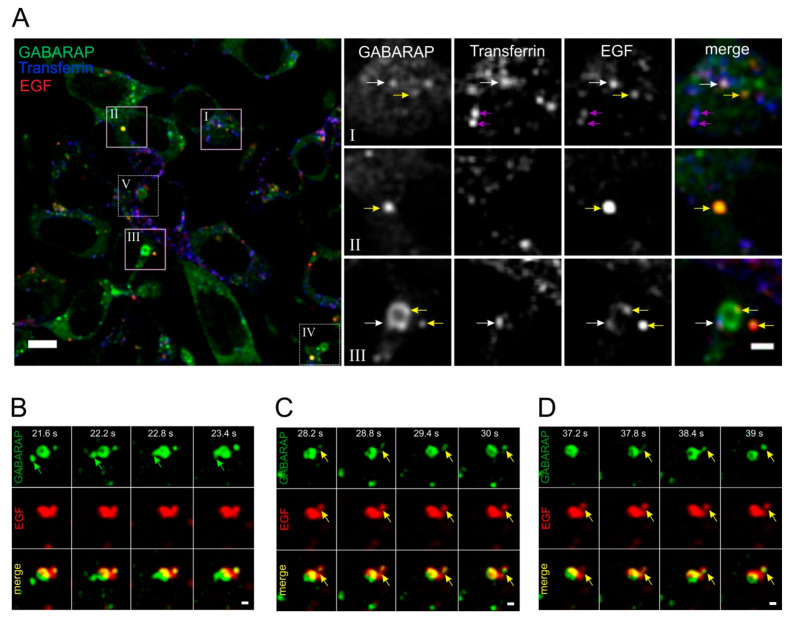
Live cell imaging of HEK293 knock-in cells expressing GFP-GABARAP under the endogenous *GABARAP* promoter after stimulation with EGF-Alexa647 and Tf-Alexa555. (**A**) HEK293 GFP-GABARAP knock-in (KI) cells were simultaneously treated with 40 ng/mL EGF-Alexa647 and 20 ng/mL Tf-Alexa555 for 60 min. and imaged under live-cell conditions by laser scanning confocal microscopy. White arrows highlight GABARAP/EGF/Tf triple-positive structures. Yellow arrows highlight GABARAP/EGF double-positive structures. Magenta arrowheads highlight EGF/Tf double-positive structures. In the merged images GABARAP fluorescence is depicted in green, EGF in red and Tf in blue. Snapshots are shown for selected time points of a 117 s time-lapse series consisting of 10 images with 13 s time intervals between images. In Appendix A, the complete time-lapse series of regions of interest I to V are shown, with a link to the corresponding movies. Scale bar in the overview = 10 µm, scale bar in close ups = 3 µm. (**B**–**D**) Spinning disk confocal fluorescence microscopy images of highly dynamic vesicles. GABARAP-only positive vesicles and large rings are highlighted by green arrows and GABARAP/EGF double-positive vesicles are highlighted by yellow arrows. In Appendix A the complete time-lapse series is shown with a link to the corresponding movie. Scale bar = 3 µm.

**Figure 8 cells-09-01296-f008:**
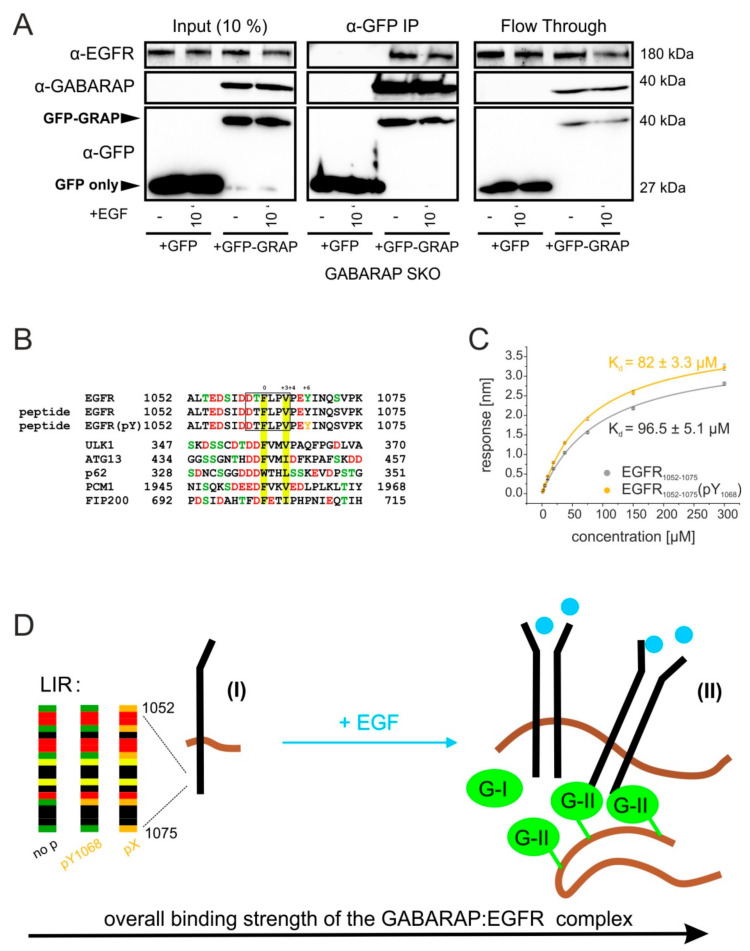
GABARAP associates with EGFR and binds to EGFR-derived peptides covering a putative LC3-interacting region (LIR) motif. (**A**) Co-Immunoprecipitation analysis between endogenous EGFR and transiently overexpressed GFP-GABARAP and GFP-only control in Huh7.5 GABARAP SKO cells. Appendix A shows uncropped source blots. GRAP = GABARAP. (**B**) Sequence alignment of residues 1052 to 1075 of the cytoplasmic domain of EGFR with LIR-peptides from known GABARAP interaction partners. The core LIR motif is boxed and aromatic and hydrophobic residues in position 0 and +3 are depicted in yellow. Residues with negative charges are shown in red. Phosphorylatable residues are depicted in green. Both peptide sequences used for the BLI measurement shown in B are also depicted. Phosphorylated residue used in modified peptide is depicted in orange. Sequences were manually aligned according to the general core consensus (W/F/Y)-X-X-(L/I/V) where × may be any amino acid. (**C**) Ascending concentrations of recombinantly expressed and purified GABARAP were titrated to immobilized peptides and response measured by BLI. Measurements were performed in triplicates. Dissociation constants (Kd) of GABARAP were 96.5 ± 5.1 µM with the unmodified EGFR LIR peptide and 82 ± 3.3 µM with the phosphorylated peptide. (**D**) Model depicting modulation of binding affinity of GABARAP and EGFR. I: Phosphorylation of aa residues in the C-terminal tail of the receptor increases binding affinity through addition of negative charges. II: Increase of local concentration of EGFR due to ligand-induced dimerization and microclustering at the plasma membrane, as well as increase of local concentration of GABARAP due to lipidation, membrane association and possibly oligomerization increase avidity and, thus, overall binding strength of the GABARAP:EGFR complex. GABARAP is represented in green. G-I = unlipidated form of GABARAP, G-II = lipidated form of GABARAP.

**Figure 9 cells-09-01296-f009:**
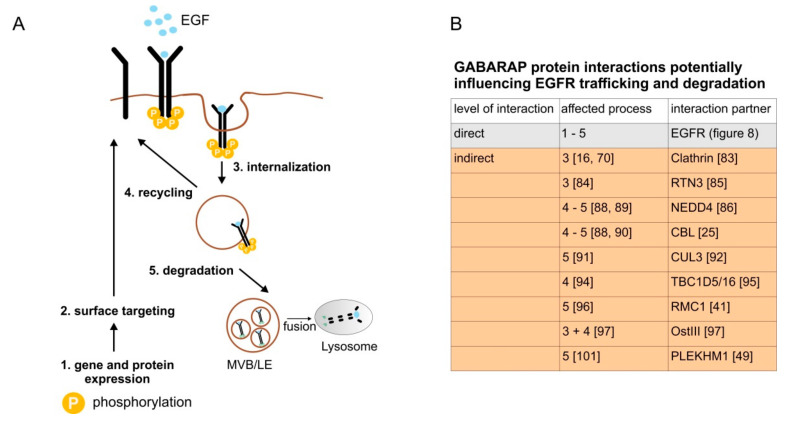
Scheme depicting EGFR internalization, trafficking and degradation including potential ways for GABARAP to take action. (**A**) 1. *EGFR* gene is expressed as mRNA and translated into protein followed by posttranslational modifications. 2. Trafficking through the Golgi apparatus regulates correct EGFR surface expression. 3. Upon extracellular ligand binding EGFR is internalized, sorted via the endosomal system and either targeted for recycling (4) or degradation (5). (**B**) List of processes that can be targeted by GABARAP either based on a direct interaction of GABARAP with EGFR as suggested in this study (grey) or indirectly by known or putative GABARAP interaction partners (light orange) with described activities in respective processes. MVB = multivesicular body, LE = late endosome.

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
