# Peer review of "Deficiency of GABARAP but Not Its Paralogs Causes Enhanced EGF-Induced EGFR Degradation"

_cells, 2020, doi:10.3390/cells9051296_

Round 1

Reviewer 1 Report

The authors use CRISPR mediated loss of function and sophisticated microscopy and cell biological analyses in two model cell lines (HEK293 and Huh7.5) to investigate the influence of the Atg8 family protein GABARAP and its two paralogs on the trafficking and degradation of EGFR in response to exogenous EGF. The authors conclude that GABARAP but not its two paralogs influence the EGF-induced trafficking and GABARAP deficiency enhances the degradation of EGFR.

The study is technically advanced, but relies on single CRISPR clones (Table S1), and lacks rescue experiment(s) that would show that the effects they report are indeed caused specifically by the lack of GABARAP and not just clonal differences. The authors should report a rescue experiment where they stably reintroduce GABARAP in to their Huh7.5 GABARAP SKO cells and show whether this has stabilised EGFR. This is the key result that needs to be added to convince me of the specificity of the effect they report.

If GABARAP influences trafficking of EGFR, would overexpression of GABARAP lead to further suppression of EGFR degradation?

If rescue experiment is not possible for some reason, the authors must at least show several clones of Huh7.5 GABARAP SKO cells that behave similarly.

Lastly, but this is not key for the validity of their results, it will be interesting to explore whether GABARAP influences trafficking induced by the low-affinity ligands of EGFR, such as amphiregulin. Note recent work showing that high-affinity ligands (such as EGF) and low-affinity ones (such as amphiregulin) lead to different signalling outcomes and different trafficking routes (e.g. Minutti et al Immunity 2019; Freed et al. Cell 2017, and references in either of them).

Reviewer 2 Report

In this manuscript, Dobner and co-workers have used CRISPR/Cas9 technology to deplete the different GABARAP subfamily members (GABARAP, GABARAPL1 and GABARAPL2) independently or together and present evidence that depletion of GABARAP enhances EGF-mediated degradation of EGFR and EGFR-dependent signaling. Neither the plasma membrane level nor the early internalization of EGFR seem to differ in GABARAP depleted cells compared to control cells. They further present evidence that inhibition of either  lysosomal or proteasomal activity restore the increased EGFR degradation seen in GABARAP deficient cells. To address the mechanisms underlying GABARAP-mediated regulation of EGFR degradation they used CRISPR/Cas9 to make knock-in cells expressing GFP-GABARAP under control of the endogenous GABARAP promoter and show very nicely using live-cell imaging that GFP-GABARAP colocalizes with internalized EGF. Finally, they identify a putative LIR (LC3 interacting region) in the cytoplasmic tail of EGFR and show by biolayer interferometry that recombinant GABARAP interacts (rather weakly) with a LIR-containing peptide.

The data presented in this manuscript are quite novel and potentially interesting, but the lack of important controls (described below) questions the conclusions made. Moreover, the data are largely descriptive with little or no mechanistic insight. The authors should address the comments below before the manuscript can be accepted for publication in Cells.

Main comments:

  1. The authors show nicely that depletion of GABARAP facilitates degradation of EGFR both in HEK239T and Huh7 cells, suggesting that a specific effect. However, only one KO clone of each is shown. In order to convincingly demonstrate that this is indeed due to lack of GABARAP, they must show that EGFR levels (as well as downstream targets such as pERK and CXCL8) are normalized upon rescue of the GABARAP SKO cell lines with GABARAP.
  2. It is unclear why the authors use overexpressed EGFR when HEK293 cells already should express endogenous EGFR. The authors find overexpressed EGFR to be degraded already under unstimulated conditions in HEK293 cells, but this is not the case for endogenous EGFR in Huh7 cells. They should also measure levels of endogenous EGFR in the HEK293 cells.
  3. Is the mRNA level of EGFR changed in GABARAP KO cells?
  4. To further strengthen their conclusion of a direct role of GABARAP in EGFR degradation, they should mutate the putative LIR in EGFR and compare the degradation of wild type versus LIR mutant EGFR in control and GABARAP SKO cells (as done in Figure 1 for WT EGFR).
  5. The GABARAP-EGFR interaction should be verified by co-immunoprecipitation of GABARAP with both WT and LIR mutant EGFR.

Minor comments:

  1. Figure 1A and 2A: should also show blotting for the respective GABARAPs. The samples included in the blots of HEK293 and Huh7 control cells should optimally be run at the same gel as the various KO cell lines for visualization of the differences between control and GABARAP KOs.
  2. Figure 1B and 2B: it would be better to normalize the values to EGFR level to T=0 min for each cell line as the graphs show the levels of transiently transfected EGFR-GFP (which could vary between cell lines) at different time points after adding EGF. It is also better to normalize the EGFR levels to T=0 min to verify that the phenotype is due to effect on degradation and not because of less expression of the protein. Why use the current y-axis scale?
  3. Figure legends in general; they should not describe the results in the legend (which is done e.g. in Fig. 1B and Fig.2B)
  4. Figure legend for figure 1: In line 332 there is a “(H)” that should be removed.
  5. Figure S1B: The cropped blots showing the verification of the DKO Huh7 cells are very small. Especially for the GABARAP L2 band in the DKO of GABARAPL1/L2, making it difficult to verify the KO. The authors should replace the blots with larger insets of the membrane.
  6. Remove the second “figure 2” in the title of figure 2 (line 387).
  7. In line 398: Move the “(C) Correlation of presence…to EGF treatment” to after the explanation of the quantification of the blots. Move to line 404, to before “Correlation analysis was…”
  8. I believe there is a typo in the abstract, line 28. “..at endogenous...” should be changed to “…as endogenous…”.
  9. In line 557, the authors write “…chloroquine treatment delayed EGF-induced EGFR degradation in Huh7 cells and restores the declined EGFR levels observed in GABARAP SKO in figure 1A”. As figure 1A shows overexpressed EGFR in HEK293A cells, the authors should rather refer to figure 2A, showing endogenous EGFR in Huh7 cells.
  10. In figure 6, it seems that chloroquine treatment partially restores EGFR levels, but not pERK. Can the authors elaborate on this?
  11. They should discuss why inhibition of the proteasome regulates EGFR degradation. They now state (line 729) that “Prior to lysosomal degradation, EGFR is deubiquitinated in a proteasome-dependent manner.” With no citation. What is the evidence for this?
  12. The discussion (and Figure 9) is confusing, too speculative and way too long.
  13. Lines 725 to 739 belongs to the figure legend of figure 9 and should be changed to the same format as the rest of the figure legend.
  14. The authors should discuss a bit more on the result showing that activation (phosphorylation) of EGFR and AKT is not affected but ERK is? And why is CXCL8 affected but not CXCL1?

Round 2

Reviewer 1 Report

The authors argue that a rescue experiment is hardly possible because overexpression of a protein or presence of an empty vector can influence the trafficking of EGFR and its degradation by influencing autophagy or aggrephagy, and CRISPR/Cas knock in is beyond the scope of this manuscript (which is probably true). If the system is so prone to perturbations, and the effects of GABARAP deficiencies on EGFR degradation rate are so small (less than 2-fold), it is remarkable that the authors are even able to measure them with statistical significance. It is surprising that the authors decline to analyse more than one independent clone of their GABARAP deficient Huh cells. It seems unlikely that they obtained only one clone per each genotype. Again, with such a sensitive system, one can easily imagine that within a population of cells, individual differences between cells of the same genotype will be of a similar magnitude as the observed effects (1.2-fold, 1.6-fold, etc.). Nevertheless, the authors instead provide alternative evidence of the specificity of the reported effects, which is analysis of endogenous EGFR in HEK cells. This shows similar trends, and support their conclusions. The updated and corrected of the manuscript has improved. It will be interesting to see whether follow-up work will validate these results and provide a molecular mechanism for the function of GABARAP and its two paralogs.

Author Response

We thank the reviewer having helped us to improve our manuscript.

Reviewer 2 Report

The authors have addressed most of my concerns, but there are still a couple missing points that need to be addressed.

  1. They have not done, or even attempted to do, the most important experiment, which is to rescue the phenotype they see in GABARAP SKO cells. They argue that this is not feasible (due to possible artificial effects of transfection/overexpression, which they do in other experiments), but this is indeed possible if one carefully selects the right system (e.g. lentiviral constructs with weak promotors allowing low expression). If this is possible, they should at least show results from more than one KO clone for each cell line.
  2. They have also not addressed further the importance of the putative LIR motif in EGFR. Although it is not possible to pull down endogenous GABARAP by co-IP, they could either try a co-IP with two overexpressed proteins (different tags) or address this with e.g. GST-GABARAP pulldown in combination with cell lysates of EGFP-EGFR wt/LIR mt. Their current data do not show a LIR-specific binding, just binding to a peptide containing a putative LIR. Thus, they cannot conclude, as they now do both in the abstract and in the heading of 3.7, that GABARAP binds to a LIR motif in the EGFR cytoplasmic tail. Without further experiments, the text must be modified.

Round 3

Reviewer 2 Report

The manuscript is now acceptable.